# IMPROVED IMAGE GENERATION VIA SPARSITY

**Roy Ganz**
Department of Electrical Engineering
Technion - Israel Institute of Technology
Technion City, Haifa 32000, Israel
ganz@cs.technion.ac.il

**Michael Elad**
Department of Computer Science
Technion - Israel Institute of Technology
Technion City, Haifa 32000, Israel
elad@cs.technion.ac.il

## ABSTRACT

The interest of the deep learning community in image synthesis has grown massively in recent years. Nowadays, deep generative methods, and specifically Generative Adversarial Networks (GANs), are leading to state-of-the-art performance, capable of synthesizing images that appear realistic. While the efforts for improving the quality of the generated images are extensive, most attempts still consider the generator part as an uncorroborated "black-box". In this paper, we aim to provide a better understanding of the image generation process. We interpret existing generators as implicitly relying on sparsity-inspired models. More specifically, we show that generators can be viewed as manifestations of the Convolutional Sparse Coding (CSC) and its Multi-Layered version (ML-CSC) synthesis processes. We leverage this observation by explicitly enforcing a sparsifying regularization on appropriately chosen activation layers in the generator and demonstrate that this leads to improved image synthesis. Furthermore, we show that the same rationale and benefits apply to generators serving inverse problems, demonstrated on the Deep Image Prior (DIP) method.

## 1 INTRODUCTION

The use of Generative Adversarial Networks (GANs) for image synthesis is one of the most fascinating outcomes of the emerging deep learning era, leading to impressive results across various generative-based tasks (Goodfellow et al., 2014; Radford et al., 2015; Zhu et al., 2017; Ledig et al., 2017; Karras et al., 2018; 2019; Brock et al., 2019). Although leading to impressive results, GANs are difficult to train and prone to undesired phenomena, such as mode collapse, failure to converge, and vanishing gradients (Thanh-Tung & Tran, 2020). Much of the research in this field has been focusing on mitigating the above difficulties and on stabilizing the training process, mainly by heuristically modifying the architectures of the generator and the discriminator, and by exposing new and better-behaved training losses (Salimans et al., 2016; Arjovsky et al., 2017; Gulrajani et al., 2017; Mao et al., 2017). As such, while GANs, in general, have been extensively studied and redesigned, the generator itself still operates as a "black-box" of unjustified architecture and meaning.

Motivated by the sparse modeling literature (Elad, 2010), we propose a novel interpretation that sheds light on the architecture of image generators and provides a meaningful and effective regularization to it. We interpret generators as implicitly relying on sparse models in general and the Convolutional Sparse Coding (CSC) and its Multi-Layered (ML-CSC) version in particular (Szlam et al., 2010; Bristow et al., 2013; Chalasani et al., 2013; Grosse et al., 2012; Heide et al., 2015; Papyan et al., 2017b; Papyan et al., 2016; Sulam et al., 2018; Sulam et al., 2018) (we provide a comprehensive overview of sparse coding in appendix B). This observation provides a possible explanation for the generator's intermediate mappings. We harness this insight by proposing a general model-based approach to regularize image generators which can be applied easily to various architectures. We validate our proposed view by conducting extensive experiments on a variety of well-known GAN architectures, from relatively simple to up-to-date ones, and show substantial performance gains.

We further extend our contribution by demonstrating that the same rationale and improvement are valid for other image generator neural networks. More specifically, we apply the proposed regularizations to the Deep Image Prior (DIP) algorithm (Ulyanov et al., 2018) for solving image denoising.

We examine the effects of our approach and show that also in this setting, in addition to image synthesis, it leads to a performance improvement.

## 2  IMPROVED IMAGE SYNTHESIS VIA GANS

Deep generative models are neural network-based architectures that synthesize signals such that their output follows the probabilistic distribution of a given data source. To this end, they map a given source distribution $P_z$ to a data distribution of interest $P_x$,

$$G(z) = x_{gen} \ \ s.t. \ z \sim P_z \ \ and \ \ x_{gen} \sim P_x. \tag{1}$$

Due to the complexity of image synthesis, the above mapping function is usually modeled by a highly expressive feed-forward deep Convolutional Neural Network (CNN), consisting of several consecutive layers. In its simplest and most common form, an image generator with $K$ layers can be rephrased as a feed-forward CNN of the form $G_K(...(G_1(z))) = x_{gen}$, where $G_i$ represents the i[th] layer of the generative model, applying convolutions, normalizations, and a non-linearity (typically ReLU). Thus, the overall mapping is attained by a sequence of $K$ transitional mappings. Despite the incremental nature of the deep image generator, it is treated as a *black-box*, without understanding the purpose of the inner activations nor enforcing a specific structure or properties on it.

In this work, we interpret existing image generators as implicitly relying on sparse modeling and propose a novel model-based approach to describe and improve the synthesis process of such architectures. Since these generators are highly expressive and over-parametrized, confining them can lead to superior results and facilitate the training process. To do so, we utilize the convolutional sparse coding (CSC) and its Multi-Layer version (ML-CSC) models. According to the CSC model, an image $x$ can be represented as a multiplication of a sparse representation vector $\Gamma$ by a convolutional dictionary $\mathbf{D}$, i.e. $x = \mathbf{D}\Gamma$. The ML-CSC further assumes that the dictionary is achieved by a multiplication of $L$ dictionaries: $\mathbf{D}_{eff} = \mathbf{D}_1\mathbf{D}_2 \cdots \mathbf{D}_L$. For additional explanation and overview regarding these models, we refer the readers to appendix B. Note, however, that although both are generative models, there is a substantial gap between their description and the process described in eq. (1). Whereas the CSC synthesis starts with a sparse representation vector, the typical image generation setup begins with a dense latent random vector $z$. To bridge this gap, we propose to interpret image generators as performing two consecutive tasks:

1. $G^S$: Map the input vector $z$ to a sparse latent vector $\Gamma$ (done by the generator's first $K - 1$ layers).
2. $G^I$: Multiply $\Gamma$ by a convolutional dictionary $\mathbf{D}$, i.e., $\mathbf{D}\Gamma \sim P_x$ (performed by the generator's K[th] layer that learns a convolutional dictionary for this purpose).

We emphasize that the second task is exactly the CSC synthesis process. By splitting the generation process into two parts, we identify the role of $\Gamma$ as the sparse representation in a (ML-)CSC-based model. This way, the image synthesis process can be described as

$$G^S(z) = \Gamma \ , \ G^I(\Gamma) = x \ \ s.t. \ z \sim P_z, \ \Gamma \ is \ sparse, \ x \sim P_x, \tag{2}$$

After establishing our view (see fig. 4 in appendix D for a visualization of our interpretation), we turn to analyze the sparsity of $\Gamma$ in regular adversarial training according to our perspective and find out that it is not sufficient (as demonstrated in fig. 3). For the purpose of designing more compatible generators with both the CSC and the ML-CSC models, we encourage $G^S$ to map $z$ to a truly sparse representation $\Gamma$. To this end, we utilize few well-known sparsifying techniques from the sparse-coding literature: (1) **$\mathbf{L_1}$ regularization** via a penalty (2) **$\mathbf{L_0}$ constraint** enforced by eliminating small non-zero entries in $\Gamma$ to satisfy a predefined sparsity constraint (3) **$\mathbf{L_{0,\infty}}$ inspired constraint** by enforcing patch-based sparsity measure (Papyan et al., 2017a; Zisselman et al., 2019), which is more compatible with the CSC variants. Although these methods induce sparsity, they are different from each other, as further explained in appendix C.

To summarize the proposed theme, we argue that learning a direct mapping from random noise to natural images' distribution is an extremely hard task that can be mitigated by enforcing a model that provides a meaningful regularization. Since the CSC model has been shown to be highly compatible with natural images, we believe that these assumptions empirically hold, and hence, the suggested approach will lead to better results.

## 3  IMPROVED SOLUTION OF INVERSE PROBLEMS

Solving inverse problems such as denoising, deblurring, inpainting, and super-resolution, is one of the most important and studied topics, and there are thousands of papers dedicated to the derivation of algorithms and strategies for handling it. In this paper, we focus on one specific and fascinating method, DIP (Ulyanov et al., 2018), which uses an image generator for handling inverse problems. According to this method, the architecture itself of convolutional deep generative models can serve as a prior for solving inverse problems. Additional overview about DIP is detailed in appendix G.

DIP is an unsupervised technique for handling inverse problems, and we find it is ideal for testing our core hypothesis – the main claim advocated in our work is that CSC-based models are adequate for generating images. If indeed correct, one would expect that CNN architectures with induced sparsity could serve better as a prior for inverse problems. Recall that both $L_0$ and $L_{0,\infty}$ sparsity constraints are enforced via an architectural modification (a projection layer). DIP puts forward an exact such test, as its essence is a reliance solely on the model's architecture itself for regularizing inverse problems. Therefore, image restoration via DIP serves as a meaningful setup for verifying our sparse-modeling assumptions. To this end, we aim to inject well-justified sparsity-inspired regularizations to specific activation layers within the generator for improving the final outcome.

Although DIP is a generic concept that can be applied to various generator architectures, we experiment with the same architecture as in the original paper – an encoder-decoder model that maps a latent vector drawn from $P_z$ into an image of the same spatial extent. Due to the significant structural difference between the encoder-decoder and standard generators, applying our method in the same way in both is unjustified. Despite this difference, a clear connection to sparse coding can be found – the blocks of the encoder part can be viewed as performing multi-layered thresholding algorithm (Papyan et al., 2016; Sulam et al., 2018; Sulam et al., 2018). Thus, we interpret the encoding process as *sparse coding* serving the ML-CSC model and enforce structural sparsity using the sparsifying techniques listed in section 2. Additional explanations regarding the connection to sparse coding and our interpretation of such architecture are provided in appendix G.

## 4  EXPERIMENTS

In this section, we experimentally examine the proposed method and compare its performance to non-regularized ("baseline") image generator architectures. First, we evaluate it on image synthesis via GANs – we conduct an extensive study on a variety of GAN architectures and explore the effect of applying our suggested regularizations to them. In addition, we examine our approach in image generation in the low data regime. Furthermore, we also show that sparsity-inducing regularization is versatile and can also be applied to more general image generators. To this end, we implement our method on the Deep Image Prior algorithm and evaluate its performance on image denoising, compared to the proper baseline. Comprehensive experiments validate that our method leads to a substantial performance improvement in both image generation and image denoising.

**Improved Image Synthesis** For experimenting with the suggested regularizations for image synthesis, we apply these on the GAN's generator during the training phase in two setups – regular and low-data regimes. In the regular-data regime setup, we conduct comprehensive experiments on various GAN architectures, conditional and unconditional, using the CIFAR-10 dataset (Krizhevsky, 2012), one of the most popular benchmarks for image synthesis. We evaluate the synthesis results with the commonly-used Fréchet Inception Distance, FID, (Heusel et al., 2017), where lower values are better, and report the results in table 1. In the low-data regime image generation, we study the effects of applying our proposed method when operating with limited datasets – we use 10% (5,000 images) and 20% (10,000 images) of the CIFAR-10 dataset. To this end, we experiment with a Big-GAN architecture trained both using the regular scheme and the differentiable augmentation method (Zhao et al., 2020), which provides state-of-the-art results on limited data. Table 2 demonstrates the performance improvement attained by applying our method in both of the setups.

These results attest that using our proposed regularization techniques significantly enhances the performance across all examined GAN models, from simple to more sophisticated up-to-date architectures, both in the regular and the low-data regime image generation. These strongly demonstrate the versatility and the generality of the proposed regularizations.

Table 1: CIFAR-10 synthesis FID results. ML corresponds to a Multi-Layer variant using $L_{0,\infty}$.

| Architecture | baseline | $\mathbf{L_{0,\infty}}$ | $\mathbf{L_0}$ | $\mathbf{L_1}$ | ML |
|---|---|---|---|---|---|
| DCGAN (Radford et al., 2015) | 37.75 | 34.45 | 35.53 | 35.52 | **32.30** |
| cGAN (Mirza & Osindero, 2014) | 27.64 | 26.43 | **26.06** | 26.48 | 26.41 |
| WGAN-GP (Gulrajani et al., 2017) | 30.06 | 28.97 | 28.51 | 30.59 | **28.04** |
| MSGAN (Mao et al., 2019) | 24.32 | **21.72** | 23.80 | 23.91 | 21.86 |
| SNGAN (Miyato et al., 2018) | 25.50 | 25.11 | 24.81 | 24.85 | **23.63** |
| SAGAN (Zhang et al., 2019) | 18.39 | 18.23 | 18.37 | 18.48 | **18.09** |
| SSGAN* (Tran et al., 2019) | 11.40 | 11.19 | 11.57 | 11.11 | **10.92** |
| BigGAN (Brock et al., 2019) | 8.23 | 7.58 | **7.50** | 7.68 | 7.66 |
| DiffCRBigGAN (Zhao et al., 2020) | 6.66 | 6.22 | 6.20 | 6.31 | **5.95** |

Table 2: Image generation results (FID) on limited data sources using BigGAN architecture.

| Dataset | BigGAN | | | | DiffCRBigGAN | | | |
|---|---|---|---|---|---|---|---|---|
| | baseline | $L_{0,\infty}$ | $L_0$ | ML | baseline | $L_{0,\infty}$ | $L_0$ | ML |
| 20% CIFAR-10 | 20.25 | 19.78 | 19.96 | **17.72** | 11.29 | 10.93 | 10.62 | **10.33** |
| 10% CIFAR-10 | 38.33 | 42.52 | **35.72** | 37.77 | 16.80 | **15.30** | 15.82 | 16.20 |

**Improved Solution of Inverse Problems** We proceed by experimenting with our regularizations on standalone image generators in the context of the DIP algorithm. Our goal is to compare the ability of regularized and non-regularized image generators to serve as an implicit prior for solving inverse problems. In the conducted experiments, we solve a denoising problem using an U-Net-like Ronneberger et al. (2015) "hourglass" architecture with skip-connections, as used in DIP (Ulyanov et al., 2018). We conduct a similar experiment to the one in DIP, using the standard denoising dataset[1]. In this setup, an observed noisy image $x_0$ is created by adding an additive white Gaussian noise (AWGN) with standard deviation $\sigma_n$ ($\sigma_n = 25$ in our experiments). To quantitatively evaluate the denoising performance we use the Peak signal-to-noise ratio (PSNR). The results are reported in table 3, where "Single" refers to the top PSNR value achieved by a single output, and "Average" is the highest PSNR value of an averaged output (obtained by an exponential sliding window over past iterations, as performed in DIP). We run each experiment multiple times to verify the statistical significance of our results. In appendix G.2, we detail the measures taken to ensure a fair comparison with the baseline.

Table 3: Denoising results of regularized and non-regularized U-Net generator using DIP.

| | baseline | $\mathbf{L_{0,\infty}}$ | $\mathbf{L_0}$ |
|---|---|---|---|
| Single | $28.74 \pm 0.03$ | $29.03 \pm 0.03$ | $\mathbf{29.22 \pm 0.06}$ |
| Average | $29.88 \pm 0.02$ | $29.94 \pm 0.04$ | $\mathbf{30.21 \pm 0.05}$ |

As can be seen, enforcing sparsity using our proposed methods outperforms the non-regularized model. These results demonstrate that the CSC modeling assumption is valid and contributes to better regularizing the inverse problem.

## 5 CONCLUSIONS

In this work, we describe simple yet effective regularization techniques for image generator architectures, which rely on sparse modeling. We demonstrate that such methods yield substantial improvements across a wide range of GAN architectures, both in regular and low-data regime setups. In addition, we show the versatility of the approach by applying it to image generators for solving inverse problems using DIP. In this context, our regularization improves the denoising results achieved by DIP. The enhanced performance achieved by promoting sparsity in image generators, in general, testifies to the relevance of sparsity-inspired models in image synthesis. We believe that this connection can be further learned and utilized to obtain even more promising results.

---

[1] http://www.cs.tut.fi/~foi/GCF-BM3D/index.html#ref_results

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

## A  Related Work

We note that a connection between sparse modeling and GANs' generators has been already proposed in Mahdizadehaghdam et al. (2019). In their work, they introduced sparsity in GANs for boosting performance and achieved their goal by designing a specific architecture that utilizes patch-based modeling and a pre-trained dictionary. While inspired by their view, our work differs substantially, as we propose a more general strategy that leverages more advanced models (CSC and ML-CSC), does not require a pre-training of any sort and can be easily applied to various existing GAN architectures. Indeed, as we shall see in the next section, our theme extends to other generators that go beyond GANs.

## B  Sparse Coding Overview

### B.1  Sparse Modeling

Sparse Modeling has been proven to be highly effective in signal and image processing applications, e.g., (Dabov et al., 2007; Bruckstein et al., 2009; Yang et al., 2010; Elad, 2010; Dong et al., 2011; 2013; Mairal et al., 2014). This model assumes an underlying linear generative model, according to which, a signal $x \in \mathbb{R}^N$ can be described as a linear combination of a few columns from a dictionary $\mathbf{D} \in \mathbb{R}^{N \times M}$, i.e. $x = \mathbf{D}\Gamma$, where $\Gamma$ is a sparse vector. The columns of $\mathbf{D}$ are referred to as atoms and they may form an overcomplete set, i.e., $M > N$. Retrieval of a sparse vector $\Gamma$, corresponding to a given a signal $x$ and a dictionary $\mathbf{D}$, is referred to as *sparse coding*, formulated as

$$\min_{\Gamma} \|\Gamma\|_0 \ s.t. \ \mathbf{D}\Gamma = x, \tag{3}$$

where $\|\Gamma\|_0$ counts the non-zeros in $\Gamma$. Thus, synthesizing a signal according to this model is done by generating a sparse representation vector $\Gamma$ and multiplying it by $\mathbf{D}$.

### B.2  Convolutional Sparse Coding (CSC)

CSC (Szlam et al., 2010; Grosse et al., 2012; Bristow et al., 2013; Chalasani et al., 2013; Heide et al., 2015; Papyan et al., 2017b) is a global variant of the above model, which is applied when handling images. The CSC has demonstrated superb performance in image processing tasks, such as denoising, separation, fusion, and super-resolution (Gu et al., 2015; Liu et al., 2016; Papyan et al., 2017a; Simon & Elad, 2019; Zisselman et al., 2019). This model's dictionary is structured, being a concatenation of banded circulant matrices containing small support filters, each appearing in all possible shifts. Thus, $\mathbf{D} \in \mathbb{R}^{N \times mN}$ where $N$ is the size of the signal $x$ and $m$ is the number of filters, each of length $n \ll N$. According to this paradigm, a signal $x$ can be expressed by $x = \mathbf{D}\Gamma$, as described above (see Figure 1).

The CSC has demonstrated superb performance in image processing tasks, such as denoising, separation, fusion, and super-resolution (Gu et al., 2015; Liu et al., 2016; Papyan et al., 2017a; Simon & Elad, 2019; Zisselman et al., 2019). A recent work (Papyan et al., 2017b) has proposed a theoretical analysis of this model, exposing the need for a redefinition of the sparsity measure to be used on $\Gamma$, in order to account for local use of atoms instead of globally counting non-zeros. We shall get back to this in the next section when using the CSC model.

Figure 1: CSC visualization: A signal $x$ is generated by a superposition of a few atoms from a convolutional dictionary $\mathbf{D}$. Each entry of $\Gamma$ corresponds to a certain shift of a limited support filter.

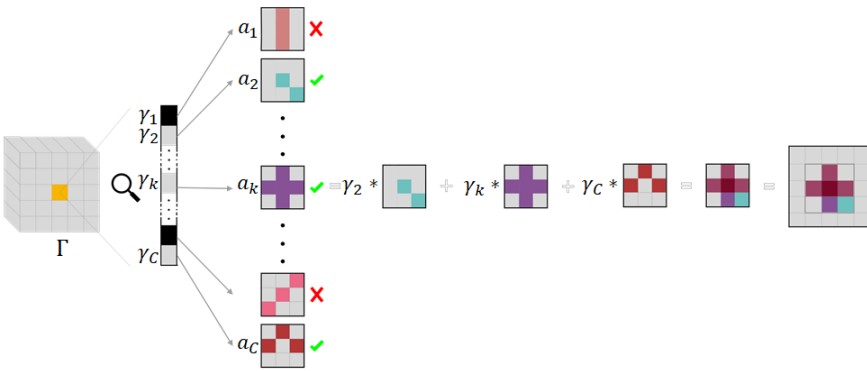

Figure 2: 2D CSC visualization: $\Gamma$ is an $H{\times}W{\times}C$ tensor, composed of $H{\times}W$ "needles", denoted as $\gamma$, each of size $1{\times}1{\times}C$. Since $\Gamma$ is sparse, the bulk of its entries is zeros. A given $\gamma$ at location $i, j$ in $\Gamma$ contributes to a patch located in the corresponding location in the output image $x$. Each non-zero entry $\gamma_i$ in $\gamma$ defines the coefficient for the filter $a_i$ in a superposition that creates a patch. Note that the patches created by nearby needles may overlap, depending on the stride and the spatial size of the atoms in $\mathbf{D}$.

### B.3 MULTI-LAYER CONVOLUTIONAL SPARSE CODING (ML-CSC)

ML-CSC (Papyan et al., 2016; Sulam et al., 2018; Sulam et al., 2018) is an extension of the CSC, which generates a cascade of sparse representations $\{\Gamma_i\}_{i=1}^L$, corresponding to CSC dictionaries $\{\mathbf{D}_i\}_{i=1}^L$. This model assumes that a signal $x \in \mathbb{R}^N$ can be represented as

$$x = \mathbf{D}_1\Gamma_1 \ \ s.t. \ \|\Gamma_1\|_0 \leq \lambda_1,$$
$$\Gamma_1 = \mathbf{D}_2\Gamma_2 \ \ s.t. \ \|\Gamma_2\|_0 \leq \lambda_2,$$
$$\vdots$$
$$\Gamma_{L-1} = \mathbf{D}_L\Gamma_L \ \ s.t. \ \|\Gamma_L\|_0 \leq \lambda_L,$$

where $\{\lambda_i\}_{i=1}^L$ are sparsity thresholds. Thus, while the first equation perfectly aligns with the regular CSC model, the additional equations add further structure by suggesting that each sparse representation vector is by itself a CSC signal. Note that by substituting the above equations, a signal $x$ can also be described as $x = \mathbf{D}_1 \cdots \mathbf{D}_i\Gamma_i = \mathbf{D}_{eff}\Gamma$, $1 \leq i \leq L$, with intermediate sparse representations.

## C SPARSITY REGULARIZATION TECHNIQUES

As stated in section 2, we examine several widely-used sparsification techniques from the sparse coding literature:

(1) **$L_1$ regularization**: Adding an $L_1$ based penalty on the representations $\Gamma$ to the overall loss of the image generator.

(2) **$L_0$ constraint**: Eliminating small non-zero entries in $\Gamma$ to satisfy a predefined sparsity constraint: $\|\Gamma\|_0 \leq \lambda$. Namely, the amount of non-zero entries in $\Gamma$ should be less or equal to $\lambda$. This can be viewed as a projection to a constraint-satisfying tensor, obtained by zeroing the smallest absolute values of the representation.

(3) **$L_{0,\infty}$ inspired constraint**: This pseudo-norm is based on a new sparsity measure related to the CSC Papyan et al. (2017b). $\|\Gamma\|_{0,\infty}$ is the maximal number of non-zero coefficients affecting any pixel in the image $x$. Thus, forcing $\|\Gamma\|_{0,\infty} \leq \lambda$ restricts the number of local atoms to $\lambda$. While this constraint is theoretically justified in the context of CSC, projection onto it is known to be challenging (Plaut & Giryes, 2019). To approximate it in a computationally plausible manner, we use a "needle"-based sparsity measure (Papyan et al., 2017a; Zisselman et al., 2019). In the general 2D CSC case, $\Gamma$ is a 3D tensor, of size $H{\times}W{\times}C$, where H and W define the image size to be synthesized, and C is the number of filters. We define a needle as $1{\times}1{\times}C$ tensor, contained in $\Gamma$.

Hence, $\Gamma$ contains $H{\times}W$ needles, each contributing to a patch of pixels in a corresponding position in the output image. In this configuration, every pixel is affected by several adjacent needles. This setup is described in Figure 2. We propose to limit the amount of non-zero entries in every such needle. To this end, for a given representation tensor $\Gamma$, we zero the smallest absolute values in each needle of the representation to satisfy the relaxed constraint.

Although the above three options all promote sparsity in $\Gamma$, they are substantially different, as demonstrated empirically in our experiments. While $L_1$ and $L_0$ consider global sparsity, the $L_{0,\infty}$ forces a local balance in the use of the atoms, i.e., limiting the local density in the representation $\Gamma$. As for the difference between $L_0$ and $L_1$, the first is deployed as a constraint, while the latter is used as a penalty. Since both $L_{0,\infty}$ and $L_0$ are constraints, we implemented them as projection layers that map an input tensor to a constraint-satisfying one by zeroing its smallest entries.

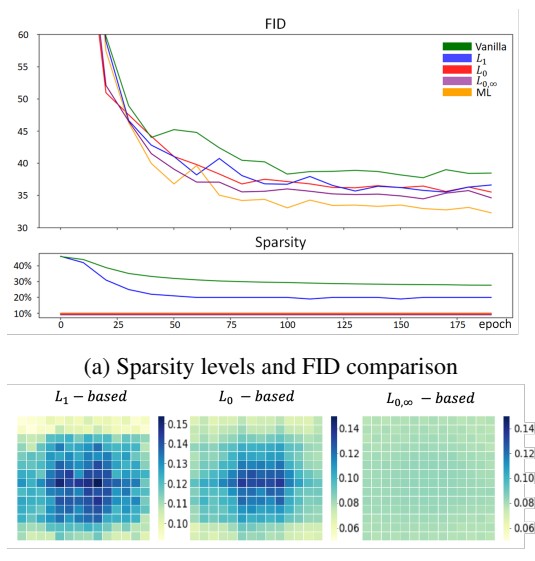

(a) Sparsity levels and FID comparison

(b) Sparsification methods' comparison

Figure 3: An illustrative experiment on the effects of applying our method during the training of a simple DCGAN (Radford et al., 2015) on the CIFAR-10 dataset. (a): Sparsity levels and FID comparison between non-regularized, $L_1$-based regularization and a constraint-based regularization ($L_0$ and $L_{0,\infty}$) on a single-layer CSC and a $L_{0,\infty}$ constraint applied on the ML-CSC. The sparsity level is the percentage of non-zeros in the representation $\Gamma$. As can be seen, promoting sparsity leads to improved performance. (b): A spatial sparsity distribution comparison of $\Gamma$, obtained by the different sparsifying techniques. Each pixel in the above figure represents the mean sparsity attained in the corresponding needle of the sparse tensor $\Gamma$. As demonstrated above, $L_0$ and $L_1$ lead to a global sparsity that is imbalanced locally, while $L_{0,\infty}$ forces such a balance. Since most of the objects in CIFAR-10 are centered, applying $L_0$ or $L_1$ regularizations leads to denser needles at the center.

## D    PROPOSED INTERPRETATION FOR IMAGE GENERATORS

We present our suggested view of convolutional image generators for synthesis purposes in section 2. According to it, the generator architecture is divided into two parts where each performs a different task – $G^S$ maps the input into a sparse representation $\Gamma$ and $G^I$ transforms $\Gamma$ into an image by multiplication with a CSC-based dictionary. A visualization of our novel view is provided in fig. 4.

## E    VISUALIZATION OF THE ML-CSC ATOMS

In the sparse coding field, it is a common practice to visualize the learned atoms in the CSC model in order to demonstrate their variety and richness. In fig. 5, we show the atoms of the ML-CSC dictionary as obtained for DCGAN trained on the CIFAR-10 database. In order to visualize the

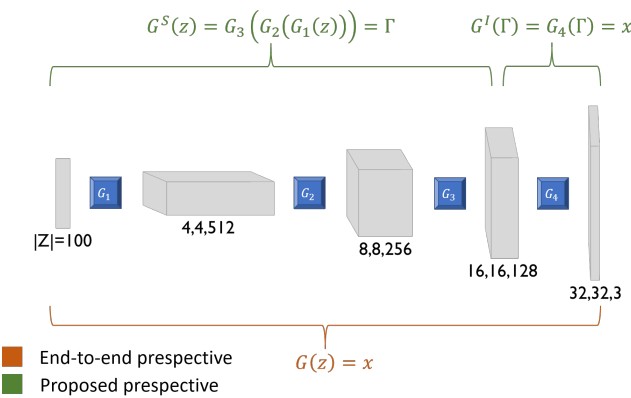

Figure 4: Our interpretation (green) of generators as handling two separate sub-tasks. The depicted architecture contains 4 blocks and the dimensions are typical for synthesizing $32 \times 32 \times 3$ images.

atoms in this two-layered ML-CSC case, we need to observe both the columns of $\mathbf{D}_1$ and $\mathbf{D}_{eff} = \mathbf{D}_1 \mathbf{D}_2$ (Papyan et al., 2016; Sulam et al., 2018). In our setup, $\mathbf{D}_1$ contains 128 atoms of size $4 \times 4 \times 3$, whereas $\mathbf{D}_2$ contains 128 atoms of size $3 \times 3 \times 128$. According to the ML-CSC, each atom in $\mathbf{D}_2$ specifies how to combine atoms from $\mathbf{D}_1$ to form an atom in $\mathbf{D}_{eff}$. Therefore $\mathbf{D}_{eff}$ contains 128 atoms, where each is of size $8 \times 8 \times 3$ and created by a sparse combination of atoms from $\mathbf{D}_1$. A visualization of the dictionaries $\mathbf{D}_1$ and $\mathbf{D}_{eff}$, trained as part of a regularized ML-CSC based DCGAN on the CIFAR-10 dataset, can be viewed in fig. 5. As expected, the atoms of $\mathbf{D}_{eff}$ are more complex than the atoms of $\mathbf{D}_1$.

## F   ADDITIONAL ANALYSIS OF SPARSITY IN GANS

We turn to present additional graphs, similar to the one given in fig. 3, that demonstrate the effect of promoting sparsity in different GAN architectures. As can be seen in fig. 6, inducing sparsity consistently improves the performance of the tested models.

## G   DEEP IMAGE PRIOR (DIP)

### G.1   BACKGROUND AND METHOD

According to DIP, a deep image generator should be trained (i.e., adapt its parameters $\theta$) to map a fixed random tensor $z$ to a given corrupted image $x_0$, and the solution to the inverse problem would be the generator's output. Formally,

$$\theta^* = \arg\min_{\theta} L(G_\theta(z)|x_0), \ \ x^* = G_{\theta^*}(z), \tag{4}$$

where $L(x|x_0)$ is a task-dependant loss term. This way, much of the information about the images' distribution $P_x$ is derived from the generator's architecture.

To better understand the proper context of sparsity in such architectures, we focus on the encoder part of the overall system. We interpret this part as transforming its input into a set of transitional representations $\{\Gamma_1, ..., \Gamma_K\}$, where $K$ is the number of scales in the architecture, as can be seen in Figure 7. These are injected into the decoder, from which it constructs the output image. As these representations are attained by a CNN, which resembles a multi-layered thresholding algorithm (Papyan et al., 2016; Sulam et al., 2018; Sulam et al., 2018), we interpret this process as *sparse coding* serving the ML-CSC model. According to this perspective, the encoder maps the input random vector to a dense signal and proceeds by performing a multi-layer *pursuit* to obtain $\{\Gamma_1, ..., \Gamma_K\}$, in the spirit of the ML-CSC model. Figure 7 shows our sparsity-related view of such architecture. Using this observation, we propose to apply our sparsity-inducing regularization, as described in section 2, on all these representations.

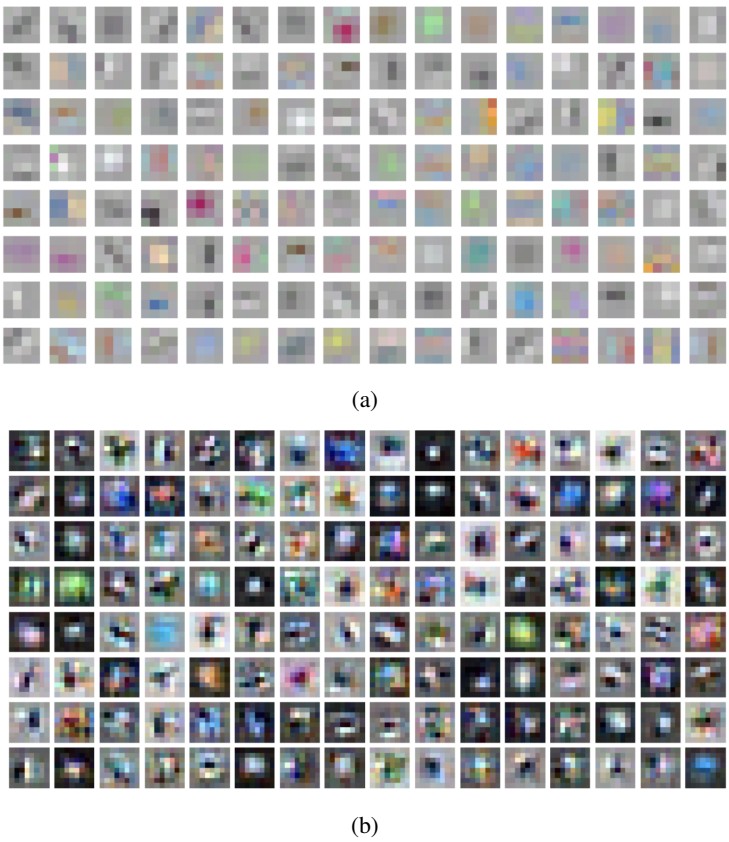

(a)

(b)

Figure 5: A visualization of the dictionary atoms in the ML-CSC setup obtained for DCGAN trained on the CIFAR-10 database: (a) The 128 atoms of $\mathbf{D}_1$, each of size $4 \times 4 \times 3$. (b) The 128 atoms of $\mathbf{D}_{eff} = \mathbf{D}_1\mathbf{D}_2$, each of size $8 \times 8 \times 3$.

## G.2 FAIR COMPARISON

In order to verify that we compare with the baseline fairly, we implement our method upon the same architecture used in DIP paper and use its provided code base. Moreover, we use the exact same hyperparameters such that the only difference between our's models and training scheme and the baseline is the sparsity regularization.

DIP algorithm is based on carefully stopping the optimization process to obtain a perceptually pleasing result. Applying our method might interfere with such a mechanism and cause a longer or shorter optimization process, thus, leading to unfair comparison. To avoid such a possibility, we consider the best image in terms of PSNR as our final prediction, both for the baseline and our approach. This way, we ensure that the only difference is in the application of sparsity regularization and that we evaluated our method adequately and fairly.

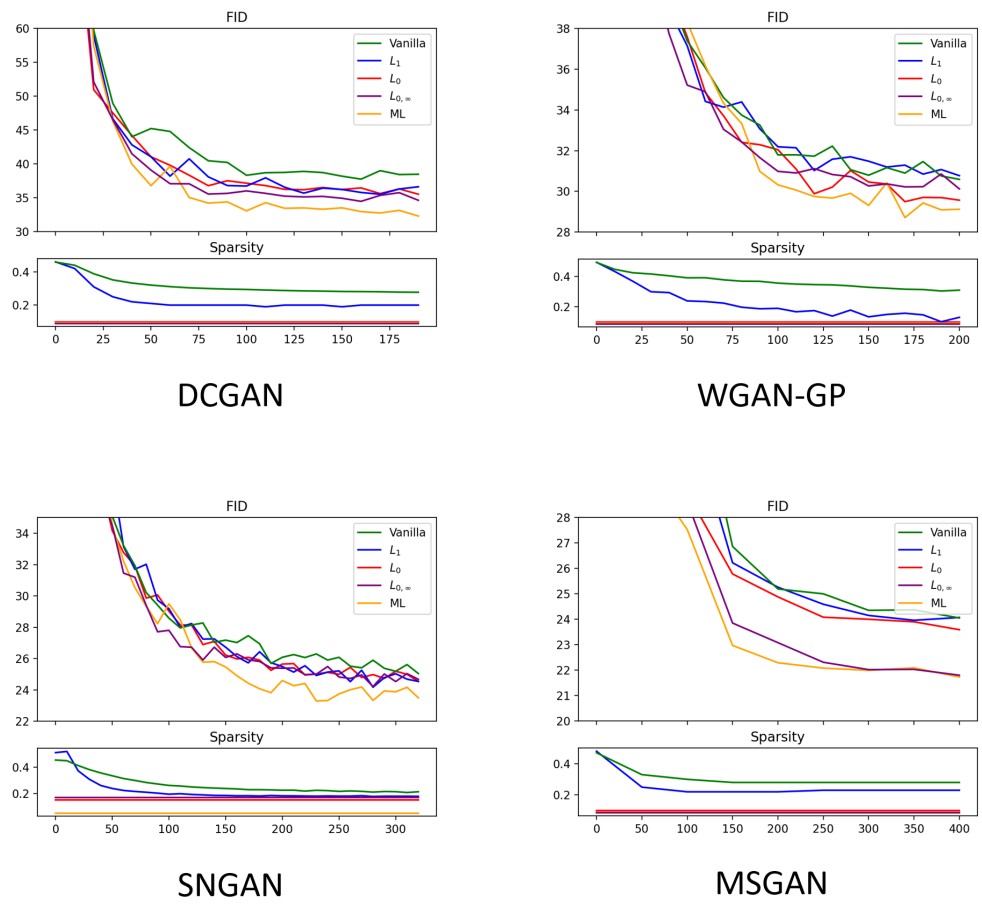

Figure 6: A visualization of the performance improvement obtained by the proposed sparsity-inducing methods across various GAN architectures.

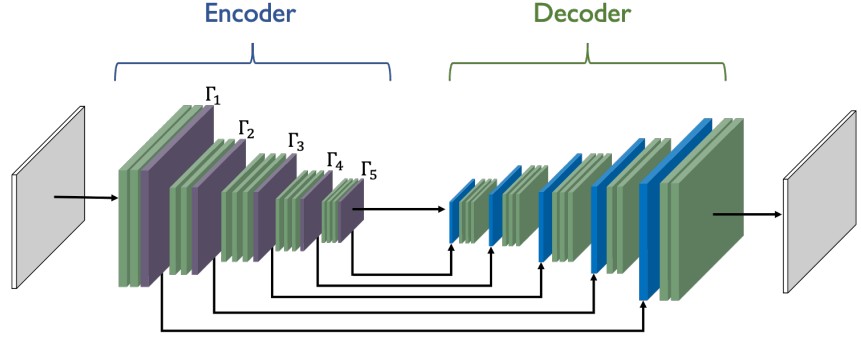

Figure 7: A visualization of our view in an encoder-decoder architecture with skip-connections.

