# OpenReview forum: "Improved Image Generation via Sparsity"
_ICLR.cc/2022/Workshop/DGM4HSD — ICLR 2022 DGM4HSD workshop Poster_

### Official Review · Reviewer_xxHh · 2022-03-13
**Interesting Empirical Findings**

**Rating:** 7
**Confidence:** 4

**Review:**

&nbsp;

## **SUMMARY**

&nbsp;

The paper proposes a regularisation scheme for deep generative models based on sparse coding. The empirical results demonstrate improved performance on the CIFAR-10 dataset across a range of GAN architectures. Future work could entail a more expansive ablation study on the sparse coding regularisation feature across a wider range of datasets and problems. I recommend acceptance with the following minor points the authors may wish to consider

&nbsp;

## **MINOR POINTS**

&nbsp;

1. A related work section in the appendix would be beneficial e.g. how the proposed method compares to [1-4].

2. Page 12 of the appendix, extra sentence, "Additional overview about DIP is detailed in appendix F."

3. Figures 3 and 6 have missing axis labels.

4. Do the authors have any rationalisation for the performance discrepancy between different sparse coding regularistion schemes in Tables 1 and 2? e.g. $L_{0,\infty}$ performs better in some experiments relative to $L_0$ and $L_1$?

5. If the authors made their code available the paper would be further strengthened.

6. On page 4, "We run each experiment multiple times to verify the statistical
significance of our results." In relation to the results given in Table 3 it would be beneficial if the exact number of trials was provided.

7. Apologies if this is an ill-posed question but the "appropriately chosen activation layers" referenced in the abstract seem to imply that there is some element of choice in applying the sparse coding regularisation scheme. Is this the case? If so is there a rationale for choosing which layers to apply the regularisation to?

&nbsp;


## **REFERENCES**

&nbsp;


[1] Mahdizadehaghdam, et al. Sparse Generative Adversarial Network. In Proceedings of the IEEE/CVF International Conference on Computer Vision Workshops 2019.

[2] Yuan et al. SparseGAN: Sparse Generative Adversarial Network for Text Generation. arXiv 2021.

[3] Zhu. et al. Tensor-Generative Adversarial Network with Two-Dimensional Sparse Coding: Application to Real-Time Indoor Localization. In 2018 IEEE International Conference on Communications (ICC) 2018.

[4] Allen-Zhu and Li. Forward super-resolution: How Can GANs Learn Hierarchical Generative Models for Real-World Distributions. arXiv 2021.

&nbsp;

---

### Official Review · Reviewer_BaQo · 2022-03-23
**Meaningful abstraction, impact uncertain**

**Rating:** 6
**Confidence:** 3

**Review:**

This paper presents a CSC / ML-CSC interpretation to generative neural networks and argues that enforcing sparse coding in the structure of image-based neural networks will help improve the quality of both generative as well as inverse networks. They implement sparse coding via $L_0$, $L_{0,\infty}$ and multi-layer sparsity regularization on activations early in the network, and demonstrate that the resulting FID scores (for generative networks) as well as PSNR scores (for restoration networks) are an improvement over non-sparse versions.

The proposed idea is interesting however similar empirical observations are seen elsewhere, e.g. in sparse autoencoders and sparse GANs. It would make the paper much better if they discuss the relationship with these previous works, and show differences if any in the performance. One result I would love to see in particular is a comparison against a DIP autoencoder where all layers (not just encoder layers) are subject to the sparsity constraint. If the latter shows much worse results than the proposed approach, it would form a good justification for the proposed abstraction as well as its use in neural networks.

I think with this added discussion the paper would be a good fit for this workshop. Additionally, this paper is well written and easy to understand.

Minor: In Table 3 caption please specify that the numbers represent PSNR values.

---

### Official Review · Reviewer_qvfY · 2022-03-28
**sparse coding and image generation**

**Rating:** 4
**Confidence:** 4

**Review:**

This paper interprets GANs or generative networks as performing sparse coding followed by multiplication by a convolutional dictionary. While this is interesting and I appreciate connections to signal processing, the analogies are somewhat vaguely stated. Missing from their analogy is dictionary learning, indeed convolutional neural networks actually learn the dictionary simultaneously with any kind of coding. Second it is not clear if this is simply an interpretation or a design suggestion, is there a clear algorithm to modify existing generators? Also the structure of the proposal can use some restructuring, sparse coding should be in the main abstract and a clear improvement methodology also stated in the main result.

---

### Decision · Program_Chairs · 2022-03-25

Accept (Poster)